# Comparison of Pathological Outcome and Recurrence Rate between En Bloc Transurethral Resection of Bladder Tumor and Conventional Transurethral Resection: A Meta-Analysis

**DOI:** 10.3390/cancers15072055

**Published:** 2023-03-30

**Authors:** Chi-Wei Wang, Ping-Jui Lee, Chih-Wei Wu, Chen-Hsun Ho

**Affiliations:** 1Department of Medicine, Division of Diabetes, Metabolism, and Endocrinology, Showa University School of MDivision of Urology, Department of Surgery, Shin Kong Wu Ho-Su Memorial Hospital, Taipei 11101, Taiwan; 2School of Medicine, College of Medicine, Fu Jen Catholic University, New Taipei City 242062, Taiwan

**Keywords:** en bloc resection of bladder tumor, detrusor muscle, non-muscle invasive bladder cancer

## Abstract

**Simple Summary:**

The conventional surgery for early-stage localized bladder tumor is transurethral resection of bladder tumor (TURBT), but this surgical method cannot preserve the integrity of tumor appearance. Therefore, it results in lower identification rate of detrusor muscle of bladder which plays an importance role in bladder cancer staging. The en bloc procedure, which has raised more concern recently, is performed by resecting the tumor completely rather than piece-by-piece. This method improved the pathological outcome and also improved the underestimation rate of bladder cancer stage. Considering its several advantages, we conducted a meta-analysis to analyze current published studies, and the results showed promising outcomes for bladder cancer staging.

**Abstract:**

Current treatment for non-muscle invasive bladder cancer (NMIBC) is the conventional transurethral resection of bladder tumor (CTURBT), but the en bloc transurethral resection of bladder tumor (ERBT) has been gaining more attraction in recent years considering better specimen integrity. Thus, we conducted this meta-analysis to compare the safety and efficacy of ERBT versus CTURBT. Trials were collected from an online database. The primary outcomes included identification of detrusor muscle in specimen, residual tumor, 3, 12, and 24-month recurrence rates and same-site recurrence rate. A total of 31 trials were included. The ERBT group had a higher rate of identification of detrusor muscle in specimens (*p* = 0.003) and lower residual tumor (*p* < 0.001). Other than that, lower rates of 3-month (*p* = 0.005) and 24-month recurrence rate (*p* < 0.001), same-site recurrence rate (*p* < 0.001) and complications were also observed. For perioperative outcomes, shorter hospitalization time (HT) (*p* < 0.001), and catheterization time (CT) (*p* < 0.001) were also revealed in the ERBT group. No significant difference was found in operative time (OT) (*p* = 0.93). The use of ERBT showed better pathological outcomes and fewer complications, so it could be considered a more effective treatment option for NMIBC.

## 1. Introduction

Bladder cancer is the 10th most common cancer worldwide and remains the most common malignancy of the urinary tract. It accounts for approximately 573,000 new cases and 213,000 deaths annually [1]. The most common clinical presentation is gross hematuria, but other symptoms also contain microscopic hematuria or irritative voiding symptoms. There are many risk factors for bladder cancer including advanced age, cigarette smoking, male sex, race, exposure to chemicals, chronic inflammatory conditions owing to infection, pelvic radiation and so on [2].

The treatment strategies for localized bladder cancer could be classified into two different phenotypes, non-muscle invasive bladder cancer (NMIBC) and muscle-invasive bladder cancer (MIBC). For this reason, the presence or absence of detrusor muscle in bladder tumor specimens reflects the quality of resection in patients with localized bladder cancer.

En bloc transurethral resection of bladder tumor (ERBT) has raised concern in recent years and has been proven to achieve good prognosis by providing a complete resection and preserving the integrity of specimens which contain detrusor muscle. It is believed that the absence of detrusor muscle in specimens was associated with residual disease, early recurrence and tumor understaging [3,4].

Since the raising concern of the en bloc surgery, more and more studies were published recently. However, there are still relatively few studies included in the existing meta-analysis for comparing these two different surgical methods. Additionally, meta-analysis which included the identification of detrusor muscle as outcome parameter were rarer. Thus, it is necessary to perform an updated meta-analysis to compare the outcomes such as different period of recurrence rate, complications and most of all, the identification of detrusor muscle. Herein, we assessed the pathological outcomes and surgical safety of ERBT and CTURBT.

## 2. Materials and Methods

The inclusion criteria were as follows: retrospective, prospective and randomized controlled trials; studies published in English language; studies that compared ERBT and conventional TURBT in the patients with NMIBC; trials that included efficacy, feasibility or pathology outcomes. The exclusion criteria in this meta-analysis included original articles published in other languages; single-arm trials, case reports, animal experiments, expert opinions, systematic reviews, conference abstracts and other meta-analyses.

We comprehensively searched the online databases such as PubMed, Cochrane Library and Embase for relevant articles published through April 2021 by using MESH terms including (bladder tumor OR bladder cancer) AND (en bloc resection) AND (transurethral resection). Overall, two authors (C.W. Wang, C.W. Wu) were involved in literature review and data extraction independently. Other than these databases, we also reviewed the previous four meta-analysis articles that compared the efficacy and feasibility of ERBT and conventional TURBT, and the articles that met our inclusion criteria collected in these meta-analysis articles but not in our literature search were included as additional records identified through other sources. After the studies were identified by the steps mentioned above, one of the authors reviewed the data from every study and assessed whether they met the criteria. The following data were extracted: surgical techniques, tumor characteristics, operation time, hospitalization time, catheterization time, 3/12/24-month recurrence rate, same-site recurrence rate, perioperative complications which included bladder irritation, bladder perforation, and obturator nerve reflex and pathological report which contained detrusor muscle in specimen and residual tumor.

The quality of the included randomized controlled trials (RCT) in our study was assessed with the Revised Cochrane risk-of-bias tool for randomized trials (RoB 2) tool [5]. On the other hand, non-randomized studies were assessed by using The Risk Of Bias In Non-randomized Studies - of Interventions (ROBINS-I) assessment tool [6]. The plots for the results above were generated with the robvis web app [7]. The parameters were divided into continuous and dichotomous variables. For the continuous one, standard mean difference with random-effect model was used to assess the difference. For the dichotomous one, the Mantel–Haenszel Test and odds ratio with 95% confidence interval and random-effect model were applied. Data analysis was performed by using Review Manager, version 5.4, and forest plot was used to evaluate the outcome between the two types of surgery.

## 3. Results

### 3.1. Studies Selection

Overall, 504 potentially relevant studies were identified. Among them, 495 studies were acquired in the primary literature search, and nine studies were included from other sources. After assessment for the eligibility, inclusion and exclusion criteria, 31 studies were finally included for data analysis. The study selection flow diagram is shown in Figure 1.

### 3.2. Characteristics of the Studies

Of all these articles, there were sixteen retrospective studies, nine prospective cohort studies and six RCTs. A total of 4195 patients were recruited; 2024 cases underwent ERBT and 2171 conventional TURBT. The characteristics of these studies including surgical methods, number of patients included in each study, sex, mean age and postoperative intravesical chemotherapy are shown in Table 1. The tumor characteristics such as size, numbers, location, stage and grade are shown in Table 2. As for surgical techniques of ERBT in these articles, electrocautery, holmium, thulium, KTP laser and hybrid knife were used; as for instilled chemotherapy, mitomycin, epirubicin, pirarubicin and BCG were used [8].

### 3.3. Tumor Complete Resection Outcome

This section contained two outcome parameters; one is identification of detrusor muscle in specimen and the other one is the residual tumor. The former one was available in 13 trials (Figure 2a). The result showed that the ERBT group had a higher identification rate of detrusor (OR 0.26; 95% CI = 0.10 to 0.63; I2 = 88%; *p* value = 0.003) and lower residual tumor (OR 0.30; 95% CI = 0.16 to 0.57, I2 = 0%; *p* value < 0.001) (Figure 2b).

### 3.4. Recurrence Related Parameters

There were four, nine and fourteen articles comprising the results of 3/12/24-month recurrence rate respectively (Figure 3a–c). The ERBT group had a relatively lower 3-month recurrence rate (Odds ratio [OR] 0.50; 95% CI = 0.30 to 0.81; I^2^ = 0%; *p* value = 0.005) and 24-month recurrence rate (OR 0.66; 95% CI = 0.52 to 0.83; I^2^ = 0%, *p* value < 0.001) as compared to the TURBT group. No difference was found in 12-month recurrence rate (OR 0.79; 95% CI = 0.49 to 1.27; I^2^ = 31%; *p* value = 0.32). As for same-site recurrence rate (Figure 3d), the ERBT had a lower rate than the TURBT group (OR 0.28; 95% CI = 0.15 to 0.52; I^2^ = 0%; *p* value < 0.001).

### 3.5. Perioperative Outcomes

Twenty-one articles contained OT (Figure 4a); the pooled mean difference [MD] was −0.08 (95% confidence interval [CI] = −1.87 to 1.72; I^2^ = 81%; *p* value = 0.93), and there was no significant difference between ERBT and TURBT. As for HT and CT, 19 and 21 articles reported the outcomes, respectively (Figure 4b,c); the pooled MD was −0.92 (95% CI = −1.28 to −0.56; I^2^ = 96%; *p* value< 0.001) and −0.77 (95% CI = −1.07 to −0.47, I^2^ = 95%, *p* value < 0.001). The ERBT group had significantly shorter HT and CT as compared to the TURBT group.

### 3.6. Complications

According to the pooled articles, the ERBT group had a significantly decreased rate of bladder perforation (OR 0.24; 95% CI = 0.13 to 0.44, I^2^ = 0%; *p* value < 0.001) (Figure 5a), obturator nerve reflex (OR 0.13; 95% CI = 0.06 to 0.29, I^2^ = 67%; *p* value < 0.001) (Figure 5b), and bladder irritation (OR 0.22; 95% CI = 0.08 to 0.60, I^2^ = 77%; *p* value = 0.003) (Figure 5c).

## 4. Discussion

In this meta-analysis, we comprehensively reviewed the past studies and provided a meta-analysis containing the most studies to compare ERBT with CTURBT for the participants with NMIBC to determine which had better pathological outcome and safety. For the perioperative outcomes, all of the three variables showed a significant test for heterogeneity, so the results would be conservative. The shorter HT and CT were found in the ERBT group and this finding was compatible with the previous meta-analysis in 2016 and 2020 [37,38]. Second, there was no significant difference in terms of 12-month RR, but ERBT showed lower 3-month, 24-month and same-site RR. Though 12-month RR showed no statistical significance, it revealed a trend that ERBT had lower 12-month RR. On the other hand, with respect to the identification of detrusor muscle in specimen, it was rarely analyzed in the previous meta-analysis owing to the small number of trials in the past. Recently, many trials were published containing the pathological outcome [12,19,23,24,28,29,30,31,32,34,35,36], so in this meta-analysis, 13 trials were enrolled, and the results were significantly superior in the ERBT compared to CTURBT group, but the test for heterogeneity was significant (*p* < 0.00001, I^2^ = 88%). Although we had this amazing result, the explanation of the result should be cautious and conservative. Furthermore, among the eight studies which showed a better rate of identification of detrusor muscle in the ERBT group, five of them performed ERBT by monopolar electrode and the others by laser fiber. It appears that a higher proportion of monopolar usage was observed, but the specific reasons are still unclear. Regarding the last primary outcome, residual tumor rate, the results show that the ERBT group had a significantly lower rate compared to the CTURBT group. Finally, ERBT showed fewer complications in bladder perforation and residual tumor and no statistical significance was noted in urethral stricture [39]. However, as for obturator nerve reflex and bladder irritation, though significant results were identified, the high heterogeneity could not be ignored. There may be some bias existing such as patients’ characteristics, demographic difference and different surgeons. Our results were compatible with the past meta-analyses; the ERBT group showed significantly better outcomes than the conventional group.

Lower incidence of obturator nerve reflex and bladder perforation was found in our study and the results were similar to the past studies. The results could be attributive to the prevention of thermal injury and the shorter contact time between the laser fiber or electrode and tumor tissue as compared to the conventional approach [26]. Herr et al. proposed three ways to assess the quality of TURBT: complete resection, deep muscle presence in the specimen and same-site recurrence rate after previous TURBT [40]. The conventional TURBT is performed by resecting the tumor piece-by-piece and it may cause incomplete tumor resection, fragmentation of the tumor tissue and also floating cancer cells. These conditions may lead to the absence of the detrusor muscle and higher tumor recurrence rate. The en bloc resection could maintain the tumor tissue integrity containing the lamina propria and detrusor muscle and enhancing the accuracy of pathological staging. By performing ERBT, we can prevent floating cancer debris and residual tumor in the tumor base and thus lower the recurrence rate, leading to a better prognosis.

The current treatment strategies for different phenotypes of bladder cancer vary greatly. NMIBC, which comprises Ta, T1 and CIS, can be treated by local resection. However, despite the new techniques introduced in this field, the postoperative recurrence rate within 12 months is up to 50% by using conventional resection [41]. Therefore, en bloc resection has caught urologists’ attention in recent years. The main difference between the two is the preservation of the tumor integrity. In the CTURBT group, the tumor is resected piece-by-piece and the tumor debris is scattered in the bladder and floated. This kind of surgical procedure violates the principle of completion resection of tumor. Furthermore, electrocautery is usually used in CTURBT, and it can cause eschar to the tumor fragments leading to difficulty in interpretation of specimen. Owing to this, the detrusor muscle is difficult to distinguish and thus it is possible to underestimate the depth of tumor invasion [29]. On the other side, the ERBT technique is performed by making an incision near the tumor down to the muscular layer and completely resecting the lesion. Recent studies have shown that it achieved good prognosis and better efficacy and safety compared with CTURBT [3].

Because of the shortcomings of the conventional TURBT, en bloc resection of bladder tumor technique has gained increasing popularity in the last decade. Theoretically, it provides a better quality of specimen by preserving the integrity of the tumor. ERBT can be performed by using eletroresection [26], laser resection or hydrodissection [31]. The procedure starts with circular coagulation on the mucosa around the tumor with a distance within 0.5–1.0 cm away from the tumor edge. Then, the incision is carried out through the submucosa layer until the detrusor muscle is exposed. The muscular fibers are cautiously dissected from the periphery to the center of the tumor with the tip of a laser fiber. The lesion is then lifted up and detached from bladder surface. After detaching completely, the tumor is retrieved by using various exit strategies. Regarding the learning curve, as most en bloc surgeries are conducted by using laser fiber, there are fewer complications such as bleeding and it provides a better surgical vision. Pankaj N. Maheshwari et al. presented a single-center study showing that the learning curve of Holmium laser ERBT is not steep and is around twenty cases [42]. Therefore, wider adoption of laser ERBT for NMIBCs is feasible and it may help to provide a better prognosis to patients with NMIBCs.

There are some limitations in this meta-analysis. First of all, this meta-analysis included retrospective studies, prospective studies and randomized controlled trials. The difference in the nature of the study types may contribute to the bias across the studies. Secondly, different techniques used in operation such as different types of lasers, electrocauterization and hybrid knife were all included in this study and as we know, laser transurethral resection has better outcomes and fewer complications than the electroresection group [10]. Thirdly, the articles collected in this meta-analysis were from the published database, PubMed, Cochrane Library and Embase, so there was no regional limitation. Thus, the patient demographics, epidemiology in different countries, and tumor characteristics can be factors that influenced the outcome. Finally, though this meta-analysis included the most articles, only six RCTs were analyzed and this has a big impact on certain outcomes. For example, as our primary endpoints, the number of RCTs which contained the result of the identification of detrusor muscle were only three articles. With such a small number, it is easy to cause bias in the results.

## 5. Conclusions

As a promising surgical technique of NMIBC, current evidence indicated ERBT is superior to CTURBT in terms of both pathological and clinical outcomes. ERBT provides a higher rate of detrusor muscle in specimen and lower rates of short-term (3 months) and intermediate-term (24 months) cancer recurrence. Furthermore, the ERBT group is associated with fewer postoperative complications, including bladder perforation and bladder irritation. However, we could not ignore the significant heterogeneity in some of these outcomes; these promising results should be explained more cautiously.

## Figures and Tables

**Figure 1 cancers-15-02055-f001:**
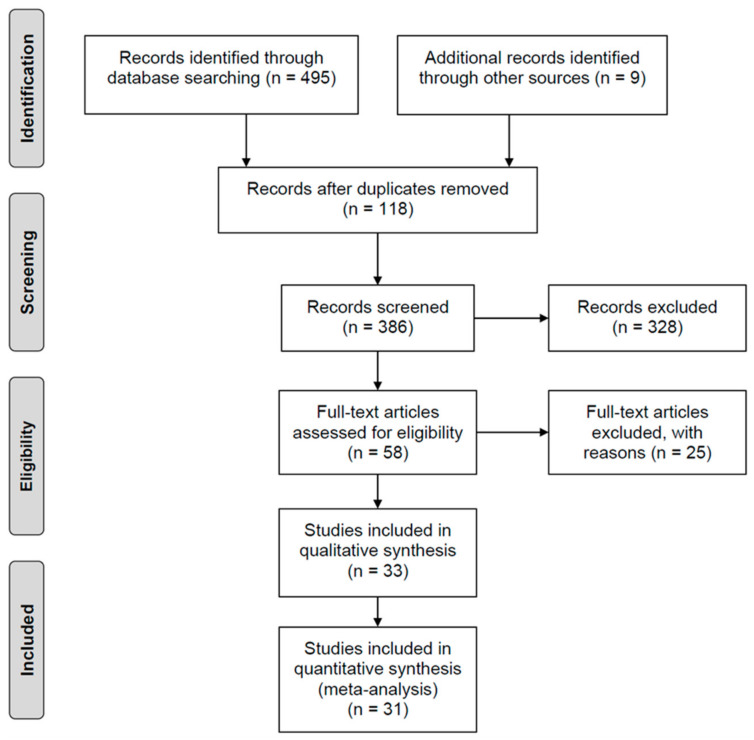
Preferred Reporting Items for Systematic Reviews and Meta-Analyses (PRISMA) flow diagram.

**Figure 2 cancers-15-02055-f002:**
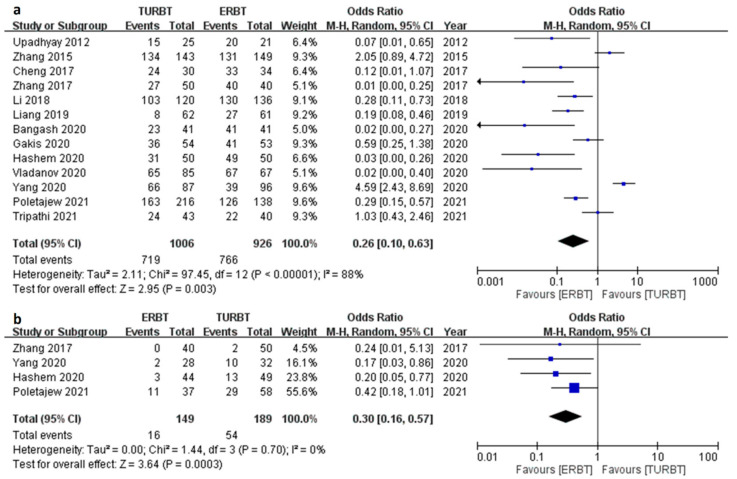
The forest plot of tumor complete resection outcome between the two groups. (**a**) Forest plot of identification of detrusor muscle. (**b**) Forest plot of residual tumor rate.

**Figure 3 cancers-15-02055-f003:**
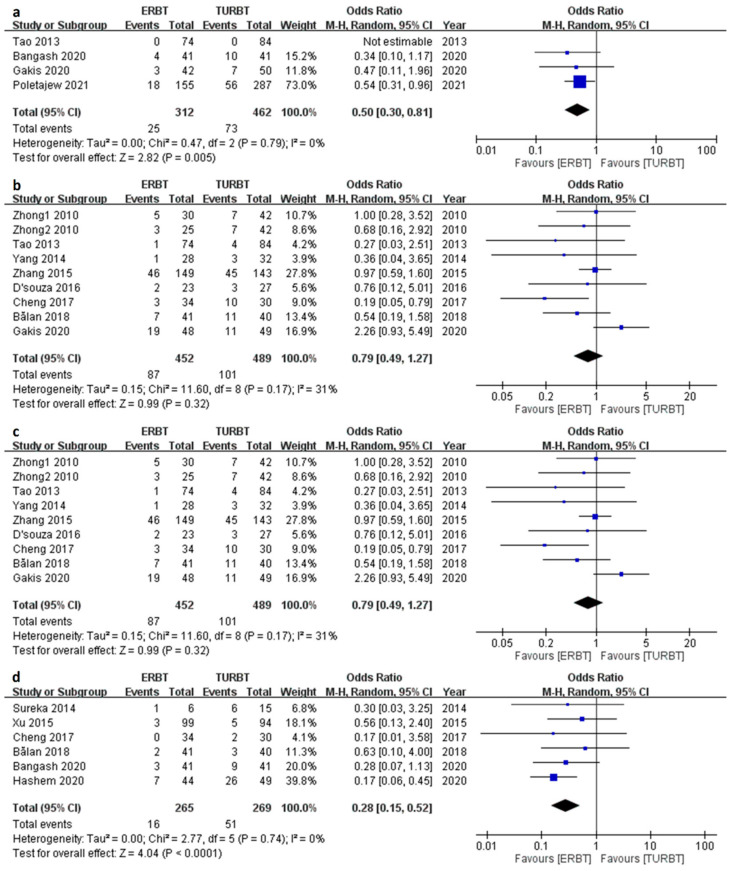
The forest plot of recurrence related parameters between the two groups. (**a**) The forest plot of 3-month recurrence rate. (**b**) The forest plot of 12-month recurrence rate. (**c**) The forest plot of 24-month recurrence rate. (**d**) The forest plot of same-site recurrence rate.

**Figure 4 cancers-15-02055-f004:**
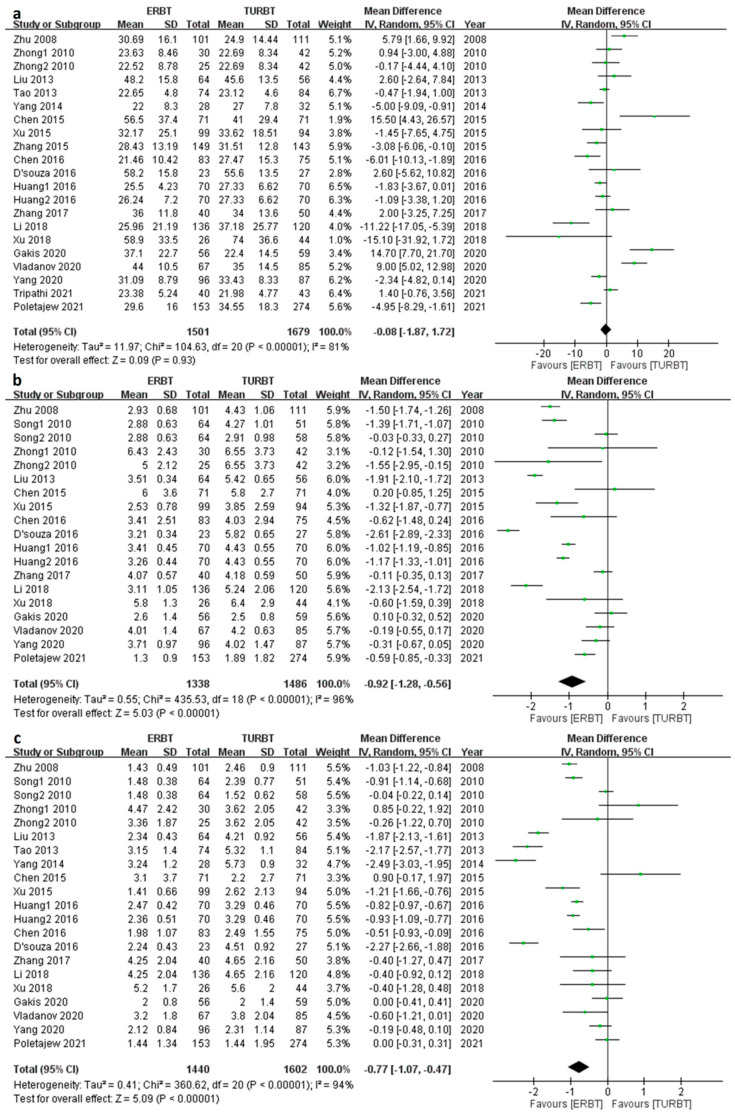
The forest plot of perioperative outcomes. (**a**) The forest plot of operation time. (**b**) The forest plot of hospitalization time. (**c**) The forest plot of catheterization time.

**Figure 5 cancers-15-02055-f005:**
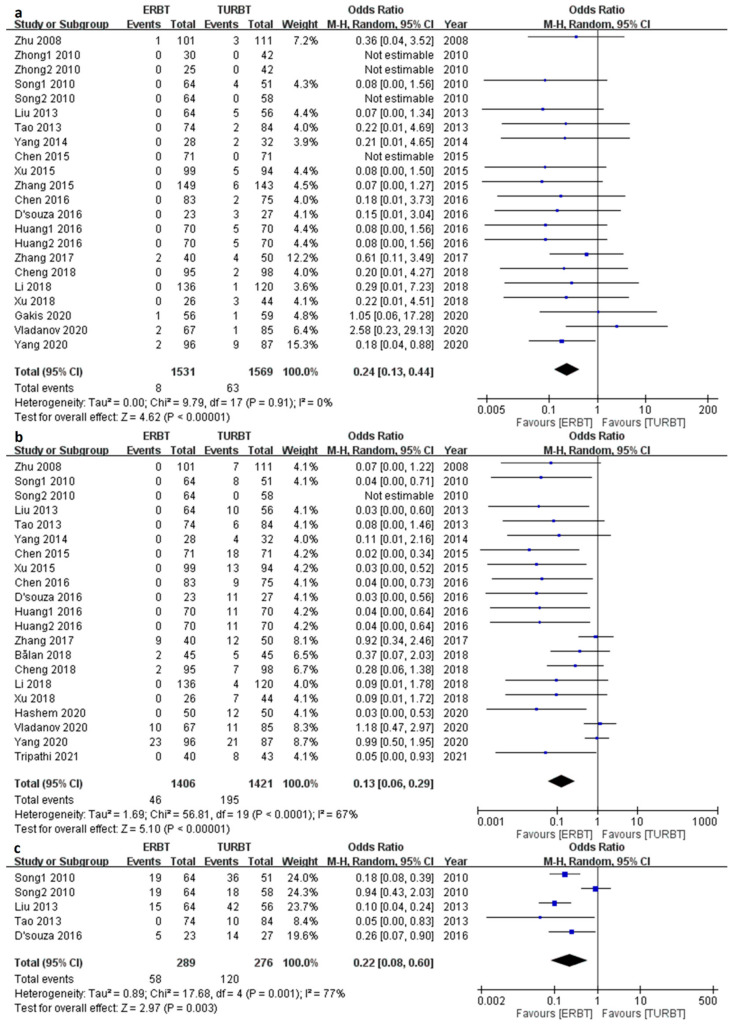
The forest plot of complications. (**a**) The forest plot of bladder perforation. (**b**) The forest plot of obturator nerve reflex. (**c**) The forest plot of bladder irritation.

**Table 1 cancers-15-02055-t001:** The general characteristics of studies included in this article. RCT, randomized controlled trial; KTP laser, Potassium-Titanyl-Phosphate laser; N/A, not available.

First Author	Year	Study Design	Surgical Method	Number of Patients		Male/Female		Mean Age		Adjuvant Therapy
			ERBT	ERBT	CTURBT	ERBT	CTURBT	ERBT	CTURBT	
Zhu et al. [9]	2008	Retrospective	Holmium laser	101	111	79/22	92/19	N/A	N/A	Mitomycin
Song1 et al. [10]	2010	Retrospective	Holmium laser	64	51	52/12	40/11	72.50	74.50	Mitomycin
Song2 et al. [10]	2010	Retrospective	Holmium laser	64	58	52/12	47/11	72.50	73.00	Mitomycin
Zhong1 et al. [11]	2010	Retrospective	Thulium laser	30	42	N/A	N/A	68.30	66.26	Epirubicin
Zhong2 et al. [11]	2010	Retrospective	Holmium laser	25	42	N/A	N/A	65.76	66.26	Epirubicin
Upadhyay [12]	2012	Prospective	Monopolar	21	25	N/A	N/A	N/A	N/A	N/A
Liu et al. [13]	2013	RCT	Thulium laser	64	56	46/18	40/16	67.10	66.30	Epirubicin
Tao et al. [14]	2013	Retrospective	KTP laser	74	84	60/14	66/18	66.4	65.3	Epirubicin
Sureka [15]	2014	Prospective	Monopolar	21	24	N/A	N/A	52.6	55	BCG
Yang et al. [16]	2014	Retrospective	KTP laser	28	32	22/6	25/7	45.3	42.5	Epirubicin
Chen et al. [17]	2015	RCT	Thulium laser	71	71	54/17	51/20	63	62	Epirubicin
Xu et al. [18]	2015	RCT	KTP laser	99	94	80/19	76/18	63.06	62.82	Pirarubicin
Zhang et al. [19]	2015	RCT	Thulium laser	149	143	70/79	79/64	N/A	N/A	Epirubicin
Chen et al. [20]	2016	Prospective	KTP laser	83	75	60/23	51/24	63.43	65.31	Mitomycin
D’souza [21]	2016	Prospective	Holmium laser	23	27	15/8	18/9	66.3	67.1	Mitomycin
Huang1 [22]	2016	Retrospective	Thulium laser	70	70	50/20	48/22	58.31	57.87	Epirubicin
Huang2 et al. [22]	2016	Retrospective	Holmium laser	70	70	45/25	48/22	59.97	57.87	Epirubicin
Cheng et al. [23]	2017	Retrospective	KTP laser	34	30	28/6	27/3	59.41	63.13	Mitomycin
Zhang et al. [24]	2017	Retrospective	Monopolar	40	50	35/5	38/12	60.65	60.8	Pirarubicin
Xu et al. [25]	2018	Retrospective	Thulium laser	26	44	24/2	35/9	55.9	59.7	Pirarubicin
Bălan et al. [26]	2018	Prospective	Bipolar	45	45	N/A	N/A	64.7	66.1	BCG + Epirubicin
Cheng et al. [27]	2018	Retrospective	Hybrid knife	95	98	67/28	70/28	62.4	64.4	Pirarubicin
Li et al. [28]	2018	Retrospective	Thulium laser	136	120	110/26	98/22	N/A	N/A	Pirarubicin
Liang et al. [29]	2019	Retrospective	KTP laser	88	70	78/10	51/19	N/A	N/A	Pirarubicin
Bangash [30]	2020	Prospective	Monopolar	41	41	34/7	36/5	58.46	58.59	Mitomycin
Gakis et al. [31]	2020	RCT	Hybrid knife	56	59	45/11	47/12	66.8	70.2	N/A
Hashem [32]	2020	RCT	Holmium laser	50	50	37/13	39/11	60.4	61.1	Epirubicin
Vladanov [33]	2020	Retrospective	Monopolar	67	85	57/10	66/19	58.43	61.5	Mitomycin
Yang et al. [34]	2020	Prospective	Monopolar	96	87	70/26	62/25	54.63	55.43	Pirarubicin
Poletajew et al. [35]	2021	Prospective	Monopolar	153	274	117/36	201/63	68	69.5	N/A
Tripathi et al. [36]	2021	Prospective	KTP laser	40	43	32/8	30/13	55.62	56.12	Mitomycin

**Table 2 cancers-15-02055-t002:** The tumor characteristics of studies included in this article. RCT, randomized controlled trial; Single, studies only enrolled the patients with single tumor; Ta noninvasive tumor; CIS, carcinoma in situ; N/A, not available. * The table shows the number of high-grade tumors, with the parentheses indicating the percentage of high-grade tumors among all tumors.

First Author	Year	Mean Tumor Size		Mean Tumor Numbers		Tumor Location				T Stage						Tumor Grade *	
						ERBT		CTURBT		ERBT			CTURBT			ERBT	CTURBT
		ERBT	CTURBT	ERBT	CTURBT	Lateral	Other	Lateral	Other	Ta	T1	CIS	Ta	T1	CIS	High Grade	High Grade
Zhu et al. [9]	2008	N/A	N/A	N/A	N/A	N/A	N/A	N/A	N/A	67	34	N/A	34	41	N/A	9(9.09%)	10 (9.01%)
Song1 et al. [10]	2010	1.85	1.74	2	1.9	25	39	20	31	36	23	5	30	17	4	20 (31.25%)	14 (27.45%)
Song2 et al. [10]	2010	1.85	1.52	2	2.2	25	39	26	32	36	23	5	35	19	4	20 (31.25%)	20 (35.09%)
Zhong1 et al. [11]	2010	2.23	1.54	1.53	1.45	N/A	N/A	N/A	N/A	23	5	2	30	8	4	5 (16.67%)	9 (21.43%)
Zhong2 et al. [11]	2010	1.38	1.54	1.4	1.45	N/A	N/A	N/A	N/A	19	5	1	30	8	4	4 (16%)	9 (21.43%)
Upadhyay et al. [12]	2012	N/A	N/A	N/A	N/A	12	9	12	13	12	6	N/A	12	8	N/A	7 (10.94%)	5 (8.93%)
Liu et al. [13]	2013	1.31	1.28	2.8	2.7	24	40	21	35	37	27	N/A	34	22	N/A	NA	NA
Tao et al. [14]	2013	N/A	N/A	1.52	1.49	62	12	69	15	50	23	1	61	21	2	23 (32.39%)	17 (23.94%)
Sureka [15]	2014	2.8	3.3	Single	Single	13	8	12	12	12	9	N/A	13	11	N/A	10 (12.05%)	12 (16%)
Yang et al. [16]	2014	N/A	N/A	N/A	N/A	22	6	17	15	8	20	N/A	7	25	N/A	3 (13.04%)	2 (7.41%)
Chen et al. [17]	2015	2.6	2.3	1.8	1.7	73	55	63	58	43	25	3	55	15	1	10 (14.29%)	6 (8.57%)
Xu et al. [18]	2015	N/A	N/A	2.16	1.91	114	100	101	79	91	8	N/A	82	12	N/A	7 (10%)	6 (8.57%)
Zhang et al. [19]	2015	N/A	N/A	N/A	N/A	N/A	N/A	N/A	N/A	106	43	N/A	107	36	N/A	25 (43.86%)	0 (0%)
Chen et al. [20]	2016	1.85	1.71	1.76	1.85	83	63	69	70	70	13	N/A	64	11	N/A	9 (22.5%)	15 (30%)
D’souza et al. [21]	2016	N/A	N/A	N/A	N/A	10	13	11	16	16	11	N/A	10	13	N/A	8 (30.77%)	13 (29.55%)
Huang1 et al. [22]	2016	1.63	1.53	2.74	2.53	28	42	25	45	40	23	7	35	27	8	NA	NA
Huang2 et al. [22]	2016	1.58	1.53	2.43	2.53	23	47	25	45	37	28	5	35	27	8	40 (42.11%)	48 (48.98%)
Cheng et al. [23]	2017	1.65	1.5	N/A	N/A	18	16	22	8	14	16	N/A	13	15	N/A	NA	NA
Zhang et al. [24]	2017	N/A	N/A	N/A	N/A	22	18	22	28	15	25	N/A	27	23	N/A	14 (34.15%)	15 (36.59%)
Xu et al. [25]	2018	N/A	N/A	N/A	N/A	21	5	30	14	10	12	N/A	25	16	N/A	9 (16.07%)	17 (28.81%)
Bălan et al. [26]	2018	1.82	1.69	N/A	N/A	N/A	N/A	N/A	N/A	24	21	N/A	23	22	N/A	16 (36.36%)	22 (44.9%)
Cheng et al. [27]	2018	2.5	2.8	N/A	N/A	N/A	N/A	N/A	N/A	52	43	N/A	54	44	N/A	NA	NA
Li et al. [28]	2018	2.39	2.15	N/A	N/A	90	68	86	58	N/A	N/A	N/A	N/A	N/A	N/A	20 (20.83%)	18 (20.69%)
Liang et al. [29]	2019	2.1	1.9	N/A	N/A	N/A	N/A	N/A	N/A	29	59	N/A	33	37	N/A	41 (29.71%)	82 (37.96%)
Bangash et al. [30]	2020	2.5	2.5	Single	Single	N/A	N/A	N/A	N/A	20	21	N/A	19	22	N/A	5 (12.50%)	6 (13.95%)
Gakis et al. [31]	2020	N/A	N/A	N/A	N/A	N/A	N/A	N/A	N/A	50	6	N/A	42	17	N/A	NA	NA
Hashem et al. [32]	2020	3.2	2.9	N/A	N/A	18	32	13	37	2	42	N/A	3	46	N/A	42 (47.73%)	26 (37.14%)
Vladanov et al. [33]	2020	N/A	N/A	N/A	N/A	38	29	41	44	35	32	N/A	49	36	N/A	5 (6.76%)	6 (7.14%)
Yang et al. [34]	2020	1.79	1.72	1.25	1.21	56	64	51	55	61	25	N/A	57	26	N/A	3 (10.71%)	6 (18.75%)
Poletajew et al. [35]	2021	N/A	N/A	N/A	N/A	N/A	N/A	N/A	N/A	94	38	21	126	58	19	10 (10.10%)	5 (5.32%)
Tripathi et al. [36]	2021	1.71	1.74	N/A	N/A	24	22	30	22	N/A	N/A	N/A	N/A	N/A	N/A	8 (5.37%)	8 (5.59%)

## Data Availability

The data presented in this study are available in this article.

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
