# Peer review of "Comparison of Pathological Outcome and Recurrence Rate between En Bloc Transurethral Resection of Bladder Tumor and Conventional Transurethral Resection: A Meta-Analysis"

_cancers, 2023, doi:10.3390/cancers15072055_

Round 1

Reviewer 1 Report

The authors have to be commended for performing a meta-analysis to compare outcomes and recurrence of CTURBT and ERBT. The manuscript is quite well organized and shows interesting results, however there are some flaws.

INTRODUCTION
- Line 39 "tracing", maybe the authors meant tract
- Line 45-47, please consider to rephrase this sentence

METHODS
- Inclusion/exclusion criteria: what about conference abstracts? Were they excluded?
- Line 82-83, "3-12-24 month recurrence rate", how was recurrence assessed?
- Wasn't performed a systematic review? The authors used the PRISMA flow-chart, nonetheless there's no mention about study registration on PROSPERO

TABLES
- tha majority of RCTs included didn't specify tumor dimension. This could bias results
- in table 2 T stage was displayed but there's no mention about tumor grade

FIGURES and RESULTS
- Figure 2a. Test for heterogeneity reveals that studies considered are not homogeneous (p<0.05), thus the pooled OR should be considered with caution
- Figure 2b. The majority of studies considered for the identification of detrusor muscle used monopolar energy. How do you interpret this result? Authors should comment
- Figure 3a showing 3-months recurrence, the study considered in the forest plot that weights more (Poletajew 2021, 73%) included only Ta-T1 tumors, no CIS, no tumor grade was showed. Please comment.
- Figures 3b-3c are the same! Please provide the correct figure 3c (24-months recurrence)
- Figure 3d, the two studies weighting more on same site recurrence used two differents surgical methods (monopolar and holmium laser). How do authors interpret this heterogeneity?
- Figure 4a, the two studies weighting more on bladder perforation used monopolar as surgical resection method. Please comment this result.
- Figure 4b-c. 
Test for heterogeneity reveals that studies considered are not homogeneous (p<0.05), thus the pooled OR should be considered with caution. Moreover, Figure 4c most weighting studies show that holmium laser + mytomycin is associated with higher rates of bladder irratation. Please comment this result.

DISCUSSION
- line 178 "shorter HT and CT", this result should be considered with caution due to the statistically significant test for heterogeneity. Same for obturatory nerve reflex and bladder irritation (lines 187-188).

Minor comments
- check that all reference are correctly listed

Reviewer 2 Report

Authors should be praised for their efforts. Nonetheless, several aspects of this manuscript need to be reviewed. 

The analysis should be done with only RCTs as it increases the statistical power. If they want to keep using the compiled analysis with the prospective/retrospective studies, they can do it on a separate analysis. 

Also, a baseline characteristic analysis should be performed. These results raised questions as to if there are differences in tumor size, stage, variant histology, etc. That can explain the results herein presented.

Round 2

Reviewer 1 Report

The authors provided a good point-to-point response to all reviewers' comments and the manuscript was improved accordingly.

Besides intrinsic biases due to the heterogeneity of studies considered and the lack of baseline tumor data, the results of the study are interesting and could form the basis for further prospective studies.

Minor comments
Consider to cite:

Lonati C, Esperto F, Scarpa RM, et al. Bladder perforation during transurethral resection of the bladder: a comprehensive algorithm for diagnosis, management and follow-up. Minerva Urol Nephrol. 2022;74(5):570-580. doi:10.23736/S2724-6051.21.04436-0

Ferro M, Barone B, Crocetto F, et al. Predictive clinico-pathological factors to identify BCG, unresponsive patients, after re-resection for T1 high grade non-muscle invasive bladder cancer. Urol Oncol. 2022;40(11):490.e13-490.e20. doi:10.1016/j.urolonc.2022.05.016

Contieri R, Hurle R, Paciotti M, et al. Accuracy of the European Association of Urology (EAU) NMIBC 2021 scoring model in predicting progression in a large cohort of HG T1 NMIBC patients treated with BCG [published online ahead of print, 2022 Oct 5]. Minerva Urol Nephrol. 2022;10.23736/S2724-6051.22.04953-9. doi:10.23736/S2724-6051.22.04953-9
